# Effects of Implementing the Timed and Targeted Counselling Model on Pregnancy Outcomes and Newborn Survival in Rural Uganda: Protocol for a Quasi-Experimental Study

**DOI:** 10.3390/mps3040073

**Published:** 2020-10-29

**Authors:** Geoffrey Babughirana, Sanne Gerards, Alex Mokori, Benon Musasizi, Nathan Isabirye, Isaac Charles Baigereza, Grace Rukanda, Emmanuel Bussaja, Stef Kremers, Jessica Gubbels

**Affiliations:** 1Department of Health Promotion, NUTRIM School of Nutrition and Translational Research in Metabolism, Maastricht University, 6211 LK Maastricht, The Netherlands; sanne.gerards@maastrichtuniversity.nl (S.G.); s.kremers@maastrichtuniversity.nl (S.K.); jessica.gubbels@maastrichtuniversity.nl (J.G.); 2Independent Scholar, Kampala 7047, Uganda; alexkmokori@gmail.com; 3World Vision International, Hoima 5319, Uganda; musasizi.benon1@gmail.com (B.M.); baigerezaisaac@gmail.com (I.C.B.); 4School of Public Health, Makerere University, Kampala 7062, Uganda; isabirye.nathan@yahoo.com; 5Lutheran World Federation, Hoima 5827, Uganda; grukanda55@gmail.com; 6Mulago National Referral Hospital, Kampala 7051, Uganda; ebussaja@gmail.com

**Keywords:** newborn survival, pregnancy outcome, household counselling, timed and targeted Counselling, uptake of services

## Abstract

Background: Although mortality rates have declined in Uganda over the last decade, maternal mortality is still high at 336 deaths per 100,000 live births, as is infant mortality at 43 deaths per 1000 live births. One in every 19 babies born in Uganda does not live to celebrate their first birthday. Many of these deaths occur within the first 28 days of life, forming the single largest category of death. Promising effects for preventing death are expected from timed and targeted counselling (ttC), an intervention package of key messages and actions that address integrated health and nutrition needs of the mothers and children, barriers and negotiation agreement, to cause sustainable behavioural change at specific timelines in the first 1000 days. Methods: The study has a quasi-experimental design in order to evaluate the implementation and effectiveness of the ttC intervention. Participants are pregnant women who have been registered by village health team (VHT) members and who live in Hoima (intervention region) or Masindi (control region) districts, who will be monitored throughout their pregnancy up to at least six weeks after delivery. A multi-stage sampling technique will be employed to select participants, the study sites being purposively chosen. Sample size is determined using the pregnancy rate from the population estimates, resulting in a total required sample of 1218 (609 each in the intervention and control group). Study instruments that will be used include the Ugandan VHT household register (in which all mothers to be studied will be registered), the ttC register (an additional tool for the study area), and a study questionnaire, to collect data at outcome level. Univariate, bivariate and multivariate analyses will be performed using SPSS to evaluate intervention effects on outcomes (e.g., relationship between pregnancy outcomes and antenatal attendance). In addition, quantitative findings will be triangulated with qualitative data, and collected through interviews and focus group discussions with participants and implementers. Discussion: The proposed study will examine the effectiveness of implementing ttC to improve maternal and child outcomes in Uganda. If ttC is effective, broader implementation of appropriate antenatal services can be advised as essential newborn care improvements. Trial registration: PACTR, PACTR202002812123868. Registered on 25 February 2020.

## 1. Background

Maternal mortality decreased by about 44% worldwide between 1990 and 2015 [1]. As part of the Sustainable Development Agenda, the current target is to reduce the global maternal mortality ratio further, to less than 70 per 100,000 live births between 2016 and 2030 [2]. Currently, every day, nearly 830 women die from preventable causes related to pregnancy and childbirth [1]. Almost all of these maternal deaths (99%) occur in developing countries, being highest among rural poor communities [3]. Within Uganda, maternal mortality declined from 527 in 1995, to 438 per 100,000 live births in 2011 and reduced further to 336 deaths per 100,000 live births in 2016 [4]. Furthermore, the Sustainable Development Goals (SDG) report [5] points out that children are most vulnerable in the first 28 days of life. By the close of the Millennium Development Goals in 2015, the global neonatal mortality rate was 19 per 1000 live births, a decrease from 31 deaths per 1000 live births in 2000. In Uganda [6], the number of infants who die before the age of 12 months has progressively decreased from 86 in 1995, to 54 in 2011, to 43 deaths [7] per 1000 live births in 2016. To further reduce these figures, there is renewed interest in the potential contribution of strengthening community health systems [8].

In Uganda, the most important reasons for the high mortality rates pertain to mobilization and support of mothers in the household regarding pregnancy and childbirth services [9]. For example, it is worrying that only 60% of pregnant women complete the recommended four antenatal care (ANC) visits, and only 29% have their first visit during the first trimester [10]. Even though 74% of deliveries in Uganda are attended by a skilled health worker, up to 44% of women and newborns do not receive postnatal care (PNC) within two days of delivery [6]. In Uganda, almost every child born has the opportunity to be breastfed (98%), yet 35% of the mothers do not initiate breastfeeding [10]. Furthermore, male involvement in ANC services in Uganda is only 65% [11]. Involving village health teams (VHTs) in maternal and newborn care can potentially go a long way to accelerate the improvement of these indicators at the community level.

The Ugandan strategy [4] of implementing the VHT at the household level aims to promote the health and wellbeing of all village members. The coverage of each VHT is at least 30 households. The role of VHTs [12] includes the mobilization of village members for health activities mainly organized by the health centres in the area, keeping village health records up to date, making monthly village health reports and organizing meetings for the village members to provide an update on the current health status for that particular village. The VHT promotes health to prevent disease, reports village sickness to health workers, performs periodic checks for danger signs in village members who are sick, and focuses on maternal newborn and child health (MNCH) services affecting behavioral change.

Timed and targeted counselling (ttC) [13] is a package of key MNCH messages and actions that are disseminated and passed on to pregnant and breastfeeding mothers, aimed at achieving sustainable behavioural change at specific time points within the first 1000 days of a child’s life. The strategy also targets key decision makers in households (e.g., husbands, grandmothers). The approach addresses integrated MNCH needs of the target groups, barriers, and negotiation of agreements with household decision makers on issues affecting the women. Targeted pregnancy outcomes refer to timely goal-oriented ANC, birth preparedness, male involvement, and delivery at a health facility, which are considered as contributing to appropriate birth outcomes. Targeted outcomes are related to a newborn survival focus on the first 28 days of life: cord care practices, essential newborn care, male involvement, and immunization as a contribution to the growth and development of the baby. The current paper describes the protocol for a study examining whether the implementation of the ttC intervention improves pregnancy outcomes and newborn survival in rural Uganda. Our hypothesis is that ttC has a positive effect on both pregnancy outcomes and newborn survival.

## 2. Methods

### 2.1. Study Design

This is a quasi-experimental study, with the implementation of ttC as the treatment condition and receiving care as usual as the control condition. The study will include both qualitative and quantitative measures to evaluate the intervention process and its effects.

### 2.2. Study Site

The study will take place in the districts of Hoima (intervention district) and Masindi (control district) in the Bunyoro sub-region in western Uganda. ttC will be implemented in the Hoima district, because this district has one of the highest maternal mortality rates in Uganda [14]. Masindi has been selected as the control district because it has a similar profile of maternal, newborn and child health determinants compared to Hoima. Some examples of similarities in key indicators for Hoima [15] and Masindi [16] are the number of females aged 18 years and above who are illiterate (38.4%, and 39.5%, respectively)**,** children aged 0–17 who lost both parents (0.7% and 0.2%), females aged 12–17 who have ever given birth (11.8% and 10.5%), and households located 5 km or more from the nearest health facility (39.5% and 29.5%).

Both Hoima and Masindi have functional VHTs who have received basic training on the roles and responsibilities in the delivery and reporting of community health. They have also received the necessary materials to assist them in their work and the two districts had VHT capacity assessed for their level of functionality before the training. All VHTs [17] in Uganda must undergo a five-day training course, be equipped with a participant’s manual, a household register, cards for counselling purposes and a monthly reporting book before they are fully qualified to provide community health services. Partners who wish to provide additional capacity-building to the VHTs first make sure that the VHTs are trained in the general strategy. However, for the intervention district, additional training has been done to facilitate the implementation of ttC to improve MNCH indicators and deliverables.

### 2.3. Ethics Approval and Consent to Participate 

Ethical approval and clearance were received from the Institutional Review Board (IRB) for the School of Public Health, Makerere University College of Health Sciences, the Higher Degrees, Research and Ethics Committee (HDREC) protocol 730 on 2 February 2020. All participants shall provide written consent to participate in the study facilitated by the VHTs.

### 2.4. Study Population

The study population for the quantitative study involves pregnant mothers registered by VHTs on suspecting the pregnancy. The study will follow these participants from pregnancy until six weeks post-delivery. For qualitative data (key informant interviews and focus group discussions), mothers shall be enrolled towards the end of the study period. Enrolment for the qualitative process evaluation will also cover VHTs, health facility staff, district health leaders and staff from the World Vision Uganda office in the department of health. To be included, women have to have either an ANC card with a child health card or a mother-child passport. Participants shall be considered discontinued if they migrate from the location included into the study at any point in time during the community health worker (CHW) follow up period.

### 2.5. Conceptual Framework

The proposed study aims to determine the effect of ttC on pregnancy outcomes and newborn survival in rural Uganda. Specifically, the study aims to ascertain the extent to which ttC improves uptake of timely goal-oriented ANC with a focus on the recommended visits, and to determine whether ttC contributes to the recommended hygienic birth practices. Additional aims are to establish the extent to which ttC improves essential newborn care practices during the newborn period, and to ascertain the extent to which ttC fosters positive male involvement in pregnancy and newborn care. Finally, the study aims to ascertain the relationship between ttC implementation and appropriate pregnancy weight gain by end of pregnancy and appropriate birth weight.

We expect that if the VHTs implement ttC effectively, there will be an increased uptake of ANC services in the intervention districts, including the timely attendance of goal-oriented ANC services, and the development and implementation of a birth plan. In addition, we expect essential newborn care improvements at the levels of the health facility and the community, aimed at newborn survival within the first 28 days of life, as well as improved visible male involvement in MNH care, including escorting the pregnant women for at least one ANC where couple human immunodeficiency virus (HIV) counselling and testing occurs. Ultimately, we expect these improvements to result in appropriate pregnancy outcomes and newborn survival (see Figure 1).

### 2.6. Intervention

The ttC intervention will be rolled out as a behavioral change communication model for the first 1000 days. For ttC to be implemented, a five-day central hands-on training is conducted for the VHTs (intermediaries), using a two-way counselling approach (participatory approach). The training will be conducted by VHT supervisors who are trained as trainers in the course for 10 days. These VHTs are then expected to translate the acquired knowledge to the expectant mothers and mothers with neonates as they deliver the ttC package during their home visits.

In the WHO’s review of global CHW training resources [18], ttC emerged as one of the most comprehensive curricula available in terms of its range of technical content, and spectrum across the continuum of care. ttC takes a life-cycle approach, focusing on supporting for the period from pregnancy up to two years of age. We believe that providing care on this continuum offers the best opportunity to ensure that children are on a path to life-long health. Taking a more family-centered approach, in ttC VHTs learn practical skills for addressing barriers to MNH, promoting involvement of fathers, and enabling CHWs to identify barriers to health [19].

VHTs are taught how to handle the household heads, any other influential individuals and the expectant mothers or mothers with neonates. During the training, the VHTs are equipped with a participant’s manual, a ttC-specific household register to note down all pregnant mothers and kick-start the timely follow-up process as scheduled in the ttC guideline [20].

The ttC visits are at given times in the first 1000 days window to pass on a given message, to trigger the household to support the mother to go for a given service at the health centre. The visits that the VHT makes are in line with the Ministry of Health (MoH) goal-oriented ANC [21] services package, the institutional delivery framework of the road to the reduction of maternal and newborn death in Uganda [22] including the essential newborn care at the facility and the community level. For this study, our focus is on the value of the four visits made during pregnancy, the three visits made when the child is in the newborn period for essential newborn care practices and a fourth visit at six weeks to enhance exclusive breastfeeding and routine immunization. The CHWs are convened by the sub county assistant on a monthly basis to ensure they adhere to the household visits and note each action in the ttC household register. The visits are specified in Table 1.

### 2.7. Delivery of Services in the Control District

In the comparison area, the VHT provide care as usual, in accordance with the reporting framework and mandate [24] of the VHT in Uganda. After performing the routine household walks and identification of the household members in a given community, the VHT is supposed to map out the households with pregnant and lactating mothers and provide support, as these visits will contribute to the task accomplishment of the VHTs as required [25]. The details of the visits is detailed in Table 2 below: 

### 2.8. Sample Size

The sample size is calculated using the formula for comparing differences in rates and proportions between two groups and determining the number of subjects per group^n^, using an individual binary for a two-sided significant level α and power 1-β [27]. With an anticipated effect size of 10% [δBinary = (^μ^T − ^μ^C)], α = 0.05 (two-sided), β = 0.2,^ z^1^−α^/2 = desired significance level at 95% (0.05). ^Ζ^1 − β = desired power at 80% (0.842), type of test = two-sided, and absolute risk reduction = 10%.

Since the study focuses on uptake of pregnancy services and essential newborn care, the key participants are pregnant mothers. We, therefore, estimate the pregnancy rate from each area using the 2020 population estimates per district, then turning the pregnancy rate into a percentage proportion. To estimate pregnancy rate, the focus will be on the recommended women of reproductive age within the population. Using the health care planning estimation, it is estimated that 24% of the population are women of reproductive age [28]. An estimation of number of births, abortions and foetal losses was acquired from the actual total 2019 health management and information systems [29] reports for both districts. This provided a 1107-sample size in the two districts. This will be increased by 10% to 1218 participants to cater for a non-response. Therefore, the treatment and control groups will each have 609 mothers at the end of the study.

To determine number of clusters needed (n_cluster) [27,30]: n _cluster = total number of participants in the two groups (/fixed cluster size) which is (^n^Cluster_Binary 2)/m = 1218/60, giving us 20 clusters. In order to have the same number of clusters per arm, the number of clusters will be increased to the nearest upper even number, which is 20 in this study. This means that the sample size contains 1218 pregnant mothers in 20 clusters which is the sub-county by the administrative units of Uganda [31].

### 2.9. Study Timelines

The study will fit into approximately two years from the time of the first participants allocation of intervention. The allocation of study timelines has been allocated in three different thematic areas of enrolment, intervention and assessments pre-and post-exposure ttC. The study period will focus on enrolment, allocation of treatment and post allocation. This will be concluded by the close out processes. The allocation of timelines in the study was assisted by the standard protocol items: recommendation for international trails (SPIRIT) timetable shown in the Table 3 below.

Women shall be enrolled in the study as they get pregnant and attend their first ANC to be able to acquire their Mothers passports and get referral to CHWs. Enrolment shall follow the stated criteria through an eligibility assessment. CHWs shall identify as many pregnant women as possible in order for the study eligibility screening to be done so that to attain the required sample. For this study, 1218 participants have to complete the interventions and therefore any drop outs before the 6 weeks after child birth will be replaced by another woman. The anticipated eligibility screening, follow up loss and eventual hitting the targeted number of participants is presented in the Consolidated Standards of Reporting Trials (CONSORT) flowchart in Figure 2 below:

### 2.10. Sampling Procedure

A multi-stage sampling technique will be employed to select study participants. A total of 20 clusters of sub-counties will be selected and included in the study, based on the power calculation. Sub-counties with largest numbers of pregnant mothers registered by the VHTs will be included to ensure that the estimated study sample size is met. This will be used to increase the level of efficiency of sampling and each cluster will be considered as a sampling unit. At the beginning of the study, the team will consult the ttC registers from the VHTs in Hoima and the VHT registers from Masindi to assess the pregnancy registration per sub-county. For each sub-county, six villages will be randomly selected, with each village providing 10 households.

At the household, selection of pregnant mothers to be included in the study will be done systematically. In this probability sampling, participants will be selected from a larger population of pregnant mothers according to a random starting point and a fixed periodic interval. This periodic interval will be determined beforehand as two, which means that assuming the starting point is one as the first serial number on the VHT register, every pregnant mother with an even serial number will be selected to participate in the study, that is, 2, 4, 6, 8, etc. The sampling procedure is presented in Figure 3.

### 2.11. Study Instruments

The study collects data from existing registries as well as through questionnaires:The Ugandan VHT household register (VHTR): This will be the entry basis into the community for both study sites. All mothers to be included in the study must have been registered by the VHT at the household level in this household register.The ttC register (ttCR): This will provide the data required for the intervention implementation analysis (process evaluation). It will also be the entry point for mothers to be included in the study and the VHTs are responsible for documentation using this register. The tool captures day-to-day visit activities between the household and the VHT.Study Questionnaire (SQ): This has been developed specifically to collect data that will be used for outcome results at the end of the intervention. An expert staff member in ttC has provided guidance that led to the revision of the tool in line with the sub-objectives. Furthermore, two researchers were asked to critically review the tool in line with the standard objectives. Their comments have been incorporated into the tool. More pre-testing of the tool will be done at the field level during the training of the enumerators.

These instruments and the variables assessed are described in more detail below.

### 2.12. Study Variables

The suggested tools above will be used to collect data on the variables in Table 4 below:

## 3. Data Management and Analysis

The data from the previous day of the study is entered and cleaned immediately, enabling immediate feedback on any issues to the data collection team. Once the team responsible for the tool returns from the field, brief conversations will be done to ensure that data was not missed at collection level.

Under the guidance of the statistician, a data entry template will be created using the expanded program of immunization info. All data shall be entered using this data entry template. Tools at data collection will be given a unique code. The process of unique coding will include sub-county, parish, village and then household number following the sampling frame. To ensure that each tool has a unique number, data collectors will be assigned numbers to use before heading into the field.

The data after entry is the responsibility of the statistician and the investigators at analysis. This data shall be attached to all publications for the world to have access to. This excludes personal data.

Univariate analysis will be done using SPSS, considering all data that requires a unilateral representation from the study. Baseline data as well as key demographic information from both comparison and implementation sites will be examined.

Several exploratory tests will be done to evaluate the data for dependencies and possible violations of normality and other key assumptions. Bivariate analysis will be used to determine the empirical relationship between study variables. This includes the differences between the implementation and comparison areas. In addition, the relation between the statuses at the baseline and the status of the indicators with a focus on the contribution of ttC will be examined. Next, multivariate analyses will be conducted, adjusting for potential relevant confounders (such as the implementation of a birth plan and a fourth ANC attendance (or more) on clean birthing practices). Depending on preliminary analyses, either fixed or random effects models will be selected for subsequent multivariate analysis, adjusting for the multi-level structure of the data, caused by the clustering of participants within villages.

### 3.1. Dissemination of Results

It is envisaged that results from this study will be shared with the community in which it has been carried out focusing on the intervention areas. The Hoima district local government, Ministry of Health and stakeholders. The World Vision Uganda Health and Nutrition Department will be getting updates on the study, and results. The results shall be published in scientific peer reviewed journals with a focus on the midterm review results and the final evaluation results. Overall, good practices will be advocated for even in those communities where the study did not take place. The Ministry of Health will be asked to replicate these findings in other parts of the country. The complete study protocol shall be available plus all data sets during the publication period.

### 3.2. Study Strength and Weaknesses

This study will have several strengths and limitations that shall be considered. The mixed methods design provided unique insights in an understudied research population. The deliberate selection of Masindi District as the control region is aimed at ensuring maximum social-demographic similarities for optimal comparability. The experimental longitudinal design will provide valuable insights, relevant for the community MNCH system policy framework.

The study anticipates to face some recruitment and retention challenges including: women in Uganda having a habit of hiding their pregnancy and therefore delaying to go for ANC and hence missing timely inclusion into the study; and limited father involvement for fear of being ridiculed by others and hence slow uptake of behaviours and services that require a male counterpart’s involvement.

The level of education for the VHTs might interfere with their ability to provide appropriate counselling for behavioural change as well as with the data collection. This is because the counselling is at times dependant on the information in the ANC card or child health card (CHC); VHTs who cannot read might not provide sufficient support to the household or register the necessary data. However, VHTs get an additional training to work with the ttC protocols, minimizing this risk. There might also be some recall bias during the end line survey, if the mothers did not manage to secure the relevant ANC and child health cards. However there is a plan to mitigate this by ensuring all pregnant women are given the cards during the first contact.

## 4. Discussion

The aim of this study is to evaluate the effect of ttC on pregnancy outcomes and newborn survival in rural Uganda with a hypothesis that ttC has no effect on pregnancy outcome and newborn survival. The backbone for this intervention is formed by the VHTs whose role it is to encourage positive behaviours for appropriate pregnancy and newborn care through community mobilization at the household level.

Pregnancy and delivery are generally considered events that do not require any medical care in most of the communities in Uganda [32]. This is due to cultural beliefs coupled with inadequate knowledge and poor household practices around maternal and newborn health care [33]. A lack of behavioral change communication to influence attitude change at the household level leads to poor and untimely pregnancy period care, with inadequate goal-oriented ANC attendance. This threatens newborn birthing practices, inadequate male partner involvement in pregnancy and newborn care, leading to poor pregnancy outcomes and newborn survival within the first 28 days of life.

Community mobilization for behavioral change such as the use of VHTs to implement ttC can be used to stimulate demand and utilisation of maternal and newborn services. In the project area, VHTs have been identified as a vehicle that can be used across the country to mobilize communities. If the VHTs are trained and equipped to implement ttC and are further supported to make timely household visits, it is hypothesized that this will affect behavioural change and therefore uptake of MNH services. ttC refers to a VHT approach of extending primary health care counselling to the household level, built around evidence-based, cost-effective key interventions for pregnant women and children under two. It is a package of key MNCH messages and actions that are disseminated and passed on to pregnant and breastfeeding mothers to bring about sustainable behavioural change at specific timelines. The strategy also targets key decision makers and those who influence decisions in households during home visits.

The study results will help in providing evidence and support programming focused on community-based support for uptake of MNCH services and influence behavioural change at the household level, through implementation of ttC. It is anticipated that if the VHTs target the mothers with key timely messages and support for positive feeding and caring practices during organized home visits, the following successes will be achieved: practices around goal-oriented ANC, essential newborn care, immediate postnatal care, and appropriate outcomes from pregnancy.

## Figures and Tables

**Figure 1 mps-03-00073-f001:**
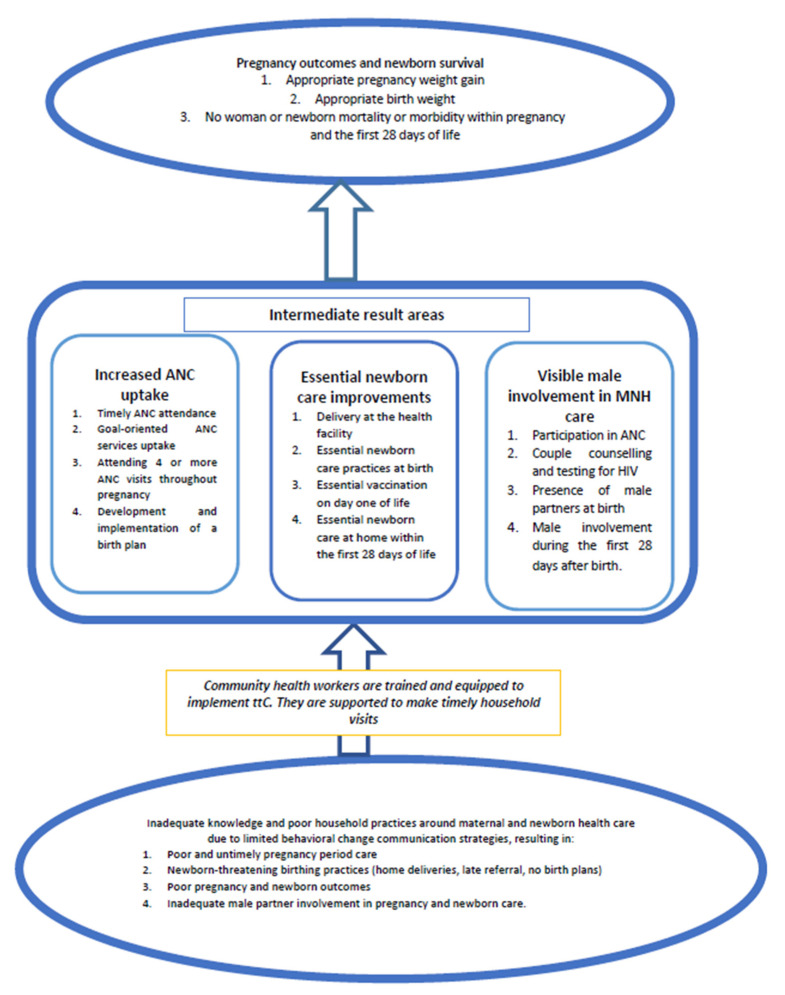
Conceptual framework. MNH—maternal and newborn health, ANC—antenatal care, ttC—timed and targeted counselling.

**Figure 2 mps-03-00073-f002:**
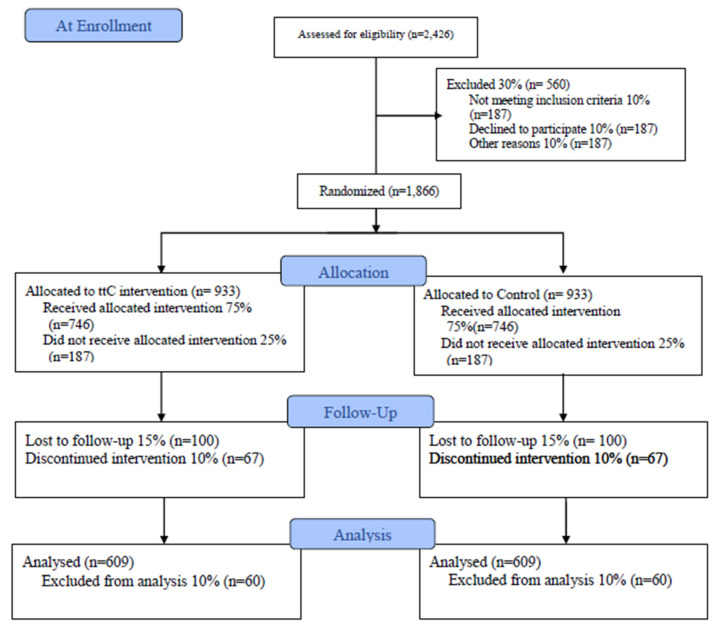
CONSORT flow diagram.

**Figure 3 mps-03-00073-f003:**
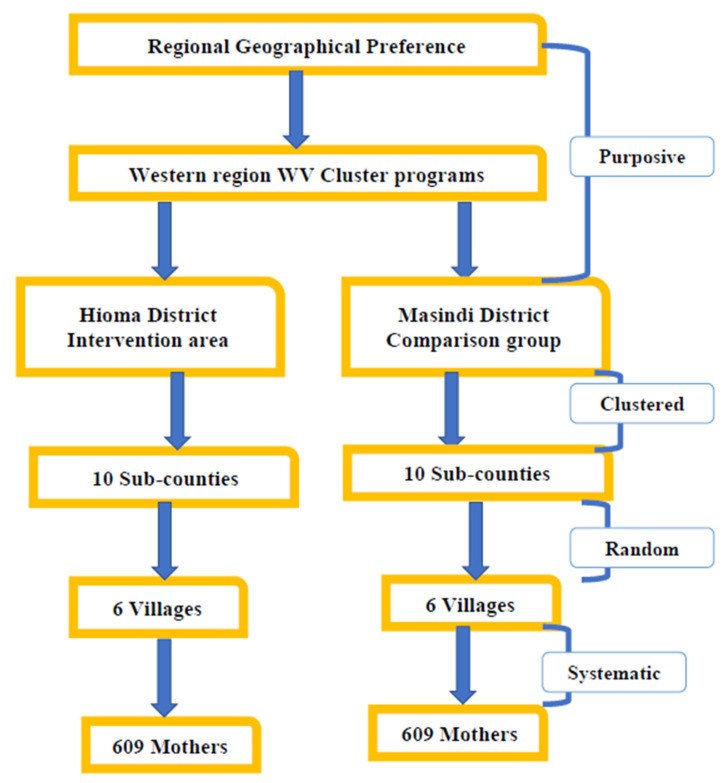
The sampling flow framework.

**Table 1 mps-03-00073-t001:** Timed and targeted counselling (ttC) visits by village health teams (VHTs) to mothers during pregnancy or after childbirth.

Visit	Timing	Services Delivered by the VHTs
1	Before or at 3 months	Focuses on suspected pregnant mothers registered as ttC candidates. They are counselled on how to care for themselves during pregnancy, a list of danger signs during pregnancy is explained, and the family is encouraged to notify the VHT in the case of any of these danger signs. Families are counselled on what to do during pregnancy and when giving birth. Women pregnant for less than three months are advised to start antenatal care at three months.
2	At 4 months of pregnancy	This is to affirm that the pregnant mother went for ANC. The pregnant women are encouraged to go for the second ANC visit. Pregnant women are educated on the advantages of exclusive breastfeeding of children until 6 months of age and continuous breastfeeding until they reach the age of 2. Advantages of giving birth in a health facility and hand washing will be highlighted during this visit.
3	At 6 months of pregnancy	Discuss the birth plan, timing and place of the birth, exploring potential challenges faced by expectant mothers in reaching the health facility and ways of overcoming these challenges. Expectant mothers are given information on the different family planning methods and where to go for services, and are encouraged to utilize one of the methods after delivery.
4	8 months of pregnancy	The mother is educated about the early signs of true labor, danger signs during labor will be explained to the households, as well as appropriate action to take in the case of these danger signs. The mother is also encouraged to deliver at a health facility and have some emergency money available. This is also the visit at which the mother needs to have a clean birthing kit available (the “Mama Kit” [23]).
5–7	The newborn period	This visit is done three times in the first week after birth, on Days 1, 3 and 7. The VHT ensures that exclusive breastfeeding and essential maternal and newborn care are not only understood but also practiced. Mothers are educated on the need to seek health care if the neonate develops fever and/or a cough, and the need to go for routine growth monitoring and immunization.
8	When the baby is one month and 2 weeks	Malaria prevention is discussed here, as well as other illnesses, danger signs in the baby and immunization for the baby and hygiene practices. The VHT checks whether the child’s growth card has been plotted, gives support for exclusive breastfeeding, checks if the child has had its HIV confirmatory test as part of the prevention of mother-to-child transition of HIV program and whether the mother has any ideas about family planning methods.

**Table 2 mps-03-00073-t002:** Visits by VHTs to mothers during pregnancy or after childbirth in the control district as part of regular care.

Visit	Timing	Services Delivered by the VHTs
ANC	Pregnancy period	Ensuring that mothers actually attend ANC as early as required to acquire the initial goal-oriented services. Making the four visits to the household with a pregnant woman.
Newborn	Newborn (28 days)	Ensuring that the cord is healing, the baby is breastfeeding and that the initial immunization has been given to the child. Ensuring that the birth took place at the health facility and—if the birth happened at home—the VHT refers the mother and baby pair to the facility.
PNC	Postnatal period	Checking on the 6th hour that the health worker gives PNC. Visit the household around the 6th day and visit around 6 weeks after delivery to check whether the mother has plans of having the last PNC visit.
Immunization	Up to 9 months after birth	Conduct household mobilization visits to check on the completeness and preparedness for child immunization [26]. The Day 1 visit is for the VHT to check on the first day immunization. The VHT visits the household at around the 6th week after birth and then at around 10 weeks and 14 weeks after delivery.

**Table 3 mps-03-00073-t003:** Study period.

Study Period								
	Enrolment	Allocation	Post-Allocation					Close-Out
**TIMEPOINT**	Feb’19	Nov’19	Jan’20	Jun’20	Jan’21	Jun’21	Jan’22	Feb’22
Enrolment								
Eligibility screen	X	X	X	X	X			
Informed consent	X	X	X	X	X			X
Allocation		X	X	X	X	X		
Interventions								
[ttC implementation]			X	X	X	X	X	
[Control implementation]			X	X	X	X	X	
Assessments								
Baseline	X	X						
MTR Outcomes						X	X	
ETR Outcome								X

MTR outcome: Midterm review outcomes focusing on results from the ttC implementation alone. ETR outcome: End term review outcomes focusing on the comparison of ttC implementation and control intervention areas.

**Table 4 mps-03-00073-t004:** Study variables.

No	Variable	Description of Variable	Tool to Be Used
Dependent Variables			
1	Appropriate pregnancy weight gain at end of months pregnancy	Pregnancy weight gain of 11.5 kg to 16 kg by the end of the pregnancy	SQ
2	Newborn morbidity within the first 28 days of life	Newborn morbidity (focusing on cord infections, sepsis, asphyxia, fever, common flu and cold and diarrhea, pneumonia) within the first 28 days of life	SQ and ttCR
3	Appropriate birth weight	Appropriate birth weight of 2.5 kg or more	SQ
4	Mortality of newborns and mothers	If the mother died during pregnancy or childbirth, if the baby died either during pregnancy, childbirth or within the first 28 days of life.	SQ and ttCR
5	Overall maternal wellness	During pregnancy or newborn period, a mother does not develop sepsis, urinary tract infections, obstruction during birth or any danger signs during this period.	SQ and ttCR
Independent Variables			
1	Uptake of timely goal-oriented ANC	With a focus on the recommended four visits, the associated recommended services offered during these visits and timeliness of attendance.	VHTR, SQ and ttCR
2	Clean birthing practices	Birth attended by a skilled health provider, delivery happened in a health facility and a mama kit was used during the process.	SQ and ttCR
3	Essential newborn care practices (ENC) during the newborn period	Early initiation of breastfeeding, cord care practices, baby thermal care, sustained breastfeeding and baby WASH practices.	SQ and ttCR
4	Positive male involvement in pregnancy and newborn care	Man participates in at least one ANC, does an HIV test with the mother, available during birth, supports mother to help her rest, contributes finances for related logistics.	SQ

Notes: SQ: Study questionnaire, ttCR: ttC register, VHTR: Ugandan VHT household register.

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
