# Peer review of "Effects of Implementing the Timed and Targeted Counselling Model on Pregnancy Outcomes and Newborn Survival in Rural Uganda: Protocol for a Quasi-Experimental Study"

_mps, 2020, doi:10.3390/mps3040073_

Round 1

Reviewer 1 Report

The protocol is well designed. The most important at this stage is to have a good plan and the ethical part taken care of. The quasi experimental approach includes both qualitative and quantitative methods, and this is well planned in the protocol.

I wish you luck with the project.

Author Response

Dear reviewer,

Thank you for your comments. We have done our best to ensure that all English errors have been corrected. infact We used an editor before submission to the journal.

Reviewer 2 Report

The topic is interesting. A good contextualization of the problem is made. The objective is clear. The methodology is adequate and realistic. The bibliographic references are correct but most of them are older than 5 years. References should be updated.

A section with the strengths and weaknesses of the project should also be included. In this section should include possible biases and how both weaknesses and biases will be addressed. In this section you should also include possible difficulties in recruiting, etc.

Author Response

Dear Reviewer,

About Your first concern about the references not being for the last 5 years before to this manuscript. Firstly, We ensured that 50% of the documents reviewed and used in the manuscript are as recent and after 2015. However its important to note that pregnancy and Newborn care in Uganda has not been studied that much and therefore there was lack of recent documents to reference. So the team expanded the search to the period of 10yrs from now. Its only a few that have a timeline of before 2010 and We think, these can be left as is.

On these comments attached is the Manuscript with tracked changes, focus on page 15, You will find an added section with weaknesses and strength and possible biases included for your consideration.

We thank You for your observation of that missing critical section.
